# Interventions and Impact of Pharmacist-Delivered Services for People Infected with COVID-19: A Systematic Review

**DOI:** 10.3390/healthcare10091630

**Published:** 2022-08-26

**Authors:** Ali Ahmed, Maria Tanveer, Sunil Shrestha, Asmita Priyadarshini Khatiwada, Saval Khanal, Juman Abdulelah Dujaili, Vibhu Paudyal

**Affiliations:** 1School of Pharmacy, Monash University Malaysia, Jalan Lagoon Selatan, Bandar Sunway 47500, Malaysia; 2Department of Pharmacy, Quaid-I-Azam University, Islamabad 45320, Pakistan; 3Department of Pharmaceutical and Health Service Research, Nepal Health Research and Innovation Foundation, Lalitpur 44700, Nepal; 4Division of Health Sciences, Warwick Medical School, University of Warwick, Coventry CV4 7AL, UK; 5School of Pharmacy, College of Medical and Dental Sciences, University of Birmingham, Birmingham B15 2TT, UK

**Keywords:** COVID-19, pharmaceutical care, medication review, physician acceptance, Pharm-D

## Abstract

Pharmacists are essential members of the healthcare team. The emergence of the novel coronavirus disease 2019 (COVID-19) pandemic has led pharmacists to undertake additional clinical roles. We aim to conduct a systematic review on the interventions and impact of pharmacist-delivered services in managing COVID-19 patients. We searched PubMed, Embase, Scopus, CINAHL plus, International Pharmaceutical Abstracts, and Web of Science from 1 December 2019 (the first case of COVID-19 emerged) to 13 January 2022 to retrieve the articles. Cochrane handbook and PRISMA guidelines were followed respectively to perform and report the review. The pharmacist-led interventions were reported following the Descriptive Elements of Pharmacist Intervention Characterization Tool (DEPICT) version 2. The protocol of systematic review was registered on PROSPERO (CRD42021277128). Studies quality was assessed with the modified NOS scale. In total, 7 observational studies were identified from 10,838 studies. Identification of dosage errors (*n* = 6 studies), regimen modifications (*n* = 5), removal of obsolete/duplicate medications (*n* = 5), identification and management of adverse drug reactions (*n* = 4), drug interactions prevention (*n* = 2), and physicians acceptance rate (*n* = 3) of therapy-related services delivered in-person or via tele-pharmacy were among the pharmacist-delivered services. Common interventions delivered by pharmacists also included optimizing the use of antibacterial, antivirals, and anticoagulants in COVID-19 infected patients. The acceptance of pharmacist-delivered services by physicians was high (88.5–95.5%). Included studies have described pharmacists’ beneficial role in managing patients with COVID-19 including detection, resolution, and prevention of medication-related problems, with physicians demonstrating high trust in pharmacists’ advice. Future research should assess the feasibility and scalability of such roles in real-world settings.

## 1. Introduction

The sudden outbreak of SARS COV-2, a coronavirus strain that caused coronavirus disease 2019 (COVID-19), significantly impacted global healthcare systems [1,2,3,4]. Managing people infected with COVID-19 requires a team of competent healthcare workers, including doctors, nurses, pharmacists, and paramedics [5,6]. Pharmacists, among other healthcare workers, are known to have played a critical role in supporting the health system during recent pandemics (tuberculosis, cholera, HIV/AIDS) by providing healthcare services to patients with chronic diseases, pharmaceutical care, telemedicine services, drugs-related information to public and health workers through their work in the community and hospital settings [7,8,9,10,11].

Pharmacists have an established role in the planning of pharmacotherapies and decisions regarding clinical parameter evaluation and drug monitoring [12,13,14,15]. Their interventions are the first line of defense in preventing medication errors (MEs) and adverse drug events (ADEs) in the first place, thereby improving medication safety [16,17,18]. Furthermore, their presence in intensive care units (ICU) and consultation with ICU physicians have been proven to minimize drug consumption, which was connected with a reduction in drug therapy cost, and to prevent inappropriate drug usage or ADEs, thereby avoiding their attributable cost [19].

As a prompt response to COVID-19, pharmacists’ professional associations, such as the International Pharmaceutical Federation (FIP), American Pharmacists Association (APhA), and the American Society of Health-System Pharmacists (ASHP), have developed guidelines for pharmacists and the pharmacy workforce to contribute to the management of the COVID-19 pandemic [20,21,22,23].

Many investigational oral antiretroviral medicines are being developed and approved for the treatment of COVID-19, and pharmacists are expected to expand pharmaceutical care activities in COVID-19 patients [24]. Several studies from high, middle, and low-income countries have reported pharmacists delivering pharmaceutical care to COVID-19 patients [6,25,26,27,28]. So far, no systematic study has been published that summarizes the expanding roles of pharmacists in COVID-19 management. Therefore, the purpose of this study was to comprehensively review and summarize the pharmacist interventions in managing medication-related problems in the care of patients with COVID-19. Health systems may want to consider involving pharmacists in the healthcare team for COVID-19 management based on the evidence from this systematic study.

## 2. Methods

We adhered to the Cochrane Handbook of Systematic Review of interventions guidelines to perform the systematic review and the Preferred Reporting Items for Systematic Reviews and Meta-Analyses (PRISMA) 2020 statement to report the review [29,30]. The protocol of this review was registered on the International Prospective Register of Systematic Reviews (PROSPERO) with the registration number CRD42021277128.

### 2.1. Information Sources and Databases Search Strategy

A comprehensive literature search was performed in PubMed, Scopus, Embase, International Pharmaceutical Abstracts (IPA) via ProQuest, Web of Science (WOS), and CINAHL Plus to retrieve data from 1st December 2019 (date of the first case of COVID-19 identified) to 13 January 2022. We also searched Google Scholar databases along with searching the citations of included studies. The search strategy comprised combinations of terms relating to COVID-19 and pharmacy. Keywords, including COVID-19, coronavirus, pharmacist, pharmacy, medication therapy management, pharmaceutical care, and medication counselling, were used. We combined these keywords using boolean operators, i.e., “OR,” “AND,” and “NOT”. In addition, we used medical subject headings “MESH” and “EMTREE” (controlled vocabulary thesaurus) for data search in PubMed and EMBASE, respectively. The search mechanism in each database was adapted and slightly modified due to technical differences and limitations. We also screened the references of the included articles and searched Google Scholar as an additional citation tracking resource to find any other studies not identified from a systematic search. Each database search keyword is given in the Appendix A. In addition, to identify studies that were not indexed in the databases listed above, a grey literature search was conducted in the Directory of Open Access Journals (DOAJ) (https://doaj.org/ accessed on 13 January 2022). No language restriction was applied in the inclusion criteria.

### 2.2. Study Selection Criteria

We used the Population, Intervention, Comparator, Outcomes, and Study Designs (PICOS) strategy in the study selection process. (Appendix A [31]. Studies were included if they included pharmacists (1) in the management of COVID-19 patients, (2) in treating COVID-19 patients independently or as a part of a multidisciplinary team (MDT), (3) in direct or indirect care or teleconsultations, and (4) if the studies were original research (randomized, cohort, or descriptive) involving patient data. We excluded qualitative studies, letters to editors, correspondences, commentaries, perspectives, and conference abstracts if not available in the full text.

### 2.3. Data Screening and Extraction

One author (AA) conducted the searches in relevant databases, which were later independently reviewed by SS and MT. Endnote version X9.3.3 (San Francisco, Clarivate Analytics) software was used to import citations of all eligible studies [32]. Subgroups were created in the Endnote software for each database, and duplicates were removed. AA, MT, and SS independently reviewed the titles and abstracts against the pre-set inclusion criteria. AA and MT did a full paper screening using a preliminary screening form, and all authors independently examined it. The final selection of articles was based on mutual agreement. After selecting the eligible studies, MT independently extracted the data using a standardized Cochrane data extraction form in a Microsoft Excel spreadsheet, and AA reviewed the extracted data for accuracy and consistency. The information extracted from the included studies were characteristics of the study population (mean age, % male, and disease state), baseline characteristics of the interventions, comparison groups, and the intervention (location, description, frequency, duration, and intensity), including primary author, objective, country, publication year, study design, and sample size. Pharmaceutical care outcomes, such as drug-related problems, therapeutic drug monitoring (TDM), ADEs reporting, and suggestions to other healthcare providers were also extracted.

### 2.4. Outcomes of Interests

The review aimed to assess the nature and outcomes of pharmacist interventions in managing COVID-19 patients. In particular, the following outcomes were considered.

(A) Primary outcomes

The number of interventions made and physician’s acceptance rate of interventions;

Drug-related problems (including identification and resolution of adverse drug reactions (ADRs), dosing errors, drug interactions, adherence to guidelines, duplication);

(B) Secondary outcomes

Quality of life (QOL), hospital readmissions rate, mortality, healthcare costs, and cost-effectiveness;

Any other relevant outcomes of pharmacist intervention as reported in the evaluations;

We excluded any study that did not measure one or more of the primary outcomes.

### 2.5. Risk of Bias

As all the included studies in this SR were observational, we used the Newcastle–Ottawa scale (NOS) to judge the risk of bias in studies [33]. NOS judges study quality based on a total of 9 stars. If a study gets 9 stars, its quality is up to the mark and fewer stars indicate a lower quality study. AA and MT made judgments, and disagreements were resolved through mutual discussions with SS.

### 2.6. Data Synthesis

The pharmacist interventions reported in the research studies were described using the Descriptive Elements of Pharmacist Intervention Characterization Tool (DEPICT) version 2 [34]. The findings of this SR are presented as a systematic narrative synthesis because meta-analysis was not feasible due to the diversity of the interventions in terms of nature, purpose, and population demographics. The details of each study’s specifics, such as its interventions and outcomes, are presented in tables and briefly discussed in the results section.

## 3. Results

A total of 10,838 potentially relevant citations were found during the electronic search. After removing duplicates and evaluating the titles and abstracts, 27 articles were selected for full-text reading. The full-text screening removed 20 articles, and only 7 articles met the inclusion criteria. Letters to the editor, commentaries, and unavailability of full text (only conference abstracts) were common reasons for excluding the studies. Furthermore, no relevant papers were found by reviewing the reference lists of the included studies. The flowchart of the literature search is represented in Figure 1. 

### 3.1. Characteristics of the Included Studies

Included studies were conducted in United Arab Emirates (UAE) (*n* = 2), United States of America (USA) (*n* = 1), China (*n* = 1), France (*n* = 1), Saudi Arabia (*n* = 1), and Thailand (*n* = 1). All studies were published in English and reported from February 2020 to January 2022. Three studies were prospective cohort studies [35,36,37], while the other four were retrospective observational studies [38,39,40,41]. Six studies were done in the inpatient setting [35,36,38,39,40,41], while one was done at a community pharmacy [37]. The studies’ sample sizes ranged from 22 [38] to 438 participants [35]. Ibrahim et al. specified 52 community pharmacies involved in the care of COVID-19 patients [35], and these selected pharmacies provided teleconsultations to probable COVID-19 patients. Besides Surapat et al. [36], all included studies mentioned the follow-up duration but none mentioned the duration of interaction with either patient or physician. Two studies did not provide any information about the age of the patients [36,37], whereas five studies did provide the mean age of patients, i.e., 46.3–68 years [35,38,39,40,41]. The participants included in the studies were pharmacists, health care providers (HCP), such as physicians, and COVID-19 positive and negative patients (part of the control group). Table 1 and Table 2 show a summary of the methods and results of the included studies. All studies described pharmacist-delivered services and evaluated the outcomes associated with pharmacist intervention.

### 3.2. Risk of Bias

Table 1 shows the overall quality scores (range 3–8 points) assigned to studies based on the research question by the authors. All studies failed to demonstrate that outcomes of interest were not present at the beginning of the study [35,36,37,38,39,40,41]. Biases in the non-exposed cohort selection [36,38,39,40,41], variations in the comparability of cohorts on a study design basis [36,38,39,40,41], and insufficient follow-up for outcomes of interest [36,38] were other reasons for the lower score. Details are provided in Appendix A.

### 3.3. Characteristics of Pharmacist-Delivered Services

Table 2 describes the pharmacist-delivered services in each study as described by DEPICT version 2 [34]. Except for Alwhaibi et al. [39], all studies performed one-to-one pharmacist contact with patients or physicians during the study duration. A total of 2825 interventions were made by pharmacists in inpatients. Six studies identified dosage errors among the five most common reasons for intervention, including overdosing and underdosing [35,37,38,39,40,41]. TDM was second of the five most common interventions, which resulted in pharmacist-initiated alterations in prescribed medication regimes [35,37,38,40,41]. Detection and prevention of ADRs were the third most common interventions [36,38,40,41] followed by the detection of duplicate drugs and their discontinuation [35,37,38,39,41], and the detection of drug–drug interactions and their management, especially in specialized populations, such as pregnant and elderly patients [37,40], as the fourth and fifth most common interventions, respectively. Pharmacists were involved in TDM in critical patients [35,36,41], monitoring electrolytes periodically, and fluid management [39]. Pharmacists participated in physical and virtual ward rounds, reviewed online electronic records, and provided virtual medication consultation and medication reconciliation in people infected with COVID-19. All the chosen studies included information on the supporting resources and materials for pharmacist actions.

### 3.4. Outcomes of Pharmacist-Delivered Services

The first significant outcome of pharmacist-delivered services was the detection of dosage errors. Parez et al. 36.7% [35], Ibrahim et al. 16.7% [37], Collins et al. 15.4% [40], Wang et al. 15.3% [42], Alwhaibi et al. 32% [39], and Al-Quteimat et al. reported 5.4% [41] of all interventions that detected incorrect or wrong dose errors, such as errors in doses of antithrombotic agents (no adjustment of heparin to renal function), antibiotics, antifungals, and antivirals. The second outcome was regime simplification, such as adjusting to antithrombotic 20.7%, antibacterial 13.8% for systemic use, and drugs for gastric acid-related disorders, 6.4% [35,38]. Parez et al. 40.9% [35], Ibrahim et al. 6.3% [37], Wang et al. 19.8% [38], Collins et al. 15.9% [40], and Al-Quteimat et al. reported 10.9% [41] interventions that resulted in regime optimization. The third outcome was the detection and prevention of ADRs. Pharmacists managed the common ADRs, such as diarrhea, body rashes, and induced hepatitis caused by protease inhibitors, azithromycin, and tocilizumab. Wang et al. 52.3% [38], Al-Quteimat et al. 18% [41], and Collins et al. reported 2.7% [40] of interventions for the prevention of ADRs. The next outcome was the detection of duplicate drugs and discontinuation of obsolete medications, e.g., the patient on Vancomycin prescribed with Linezolid [39]. A total of 7.7%, 11.7%, and 4.3% of interventions done by Ibrahim et al. [37], Alwhaibi et al. [39], and Al-Quteimat et al. [41], respectively identified and solved the duplicate errors. Al-Quteimat et al. reported 3.9% [41], Wang et al. reported 31.5% [38], and Parez et al. reported 34% [35] of interventions that led to discontinuation of the drugs. Reduction in potential drug interactions was another outcome of pharmacist intervention, as shown in the studies by Ibrahim et al. 20.8% [37] and Collins et al. 2% [40]. Collins et al. reported that 9.9% of interventions focused on COVID-19 prophylaxis, 3.6% of interventions helped balance electrolytes, and 3.2% of interventions were concerned with monitoring body fluid management in COVID-19 patients [40]. Three studies reported that physicians accepted pharmacist interventions at a rate ranging from 88.5% to 95.5% [35,38,41]. The detailed outcomes of the included studies are mentioned in Table 3.

## 4. Discussion

There is growing evidence that the SARS COV-2 infects both the upper and lower respiratory tracts with frequent multi-organ impacts, blood clots, and an unusual immune-inflammatory response not commonly associated with similar viruses [43,44]. For its treatment, various multi-therapy approaches are used in patients admitted to critical care units of hospitals, resultantly a greater risk of medication errors. This SR provides examples of pharmacist-delivered services in the treatment of COVID-19 patients to avoid medication-related errors. To the authors’ knowledge, this is the first review that synthesized evidence of pharmacist-delivered services in managing patients with COVID-19. In COVID-19 patients, pharmacist interventions identified preventable ADRs, dose and dosage errors, drug interactions and prescribing errors. Interventions were helpful in managing ADRs, avoiding drug interactions, recommending drug substitutes, and adjusting doses of antithrombotic, antibacterial, and antifungal medications. This demonstrates that incorporating pharmacists as a member of the MDT in COVID-19 management can promote better pharmaceutical care in collaboration with the nurse and physician.

In all studies, pharmacists were graduated, licensed, and trained in infectious diseases, and their knowledge of COVID-19 was continuously updated [35,36,37,38,39,40,41]. Pharmacists performed 2825 interventions, and physicians accepted more than 90% of the interventions [35,36,38,39,40,41]. Pharmacists collected medication-related issues from patient files and online records, analyzed them and then provided educational materials and solutions to patients and physicians. Since gatherings were not allowed because of the pandemic’s safety measures, educational materials/interventions were delivered to the physician in the form of pamphlets, and posters with all kinds of observed medication errors hung in various departments and prescribers’ rooms [35,36,37,38,39,40,41]. On-site education and verbal communication through phone calls were carried out. Moreover, online educational meetings for sharing experiences, online weekly educational meetings with clinicians, and the distribution of clinical protocols and guidelines were also done [37,41].

Common services delivered by the pharmacists reported in studies targeted several components of pharmaceutical care in COVID-19 patients, such as identification of drug-related problems, incorrect dosage, under/overdosage, drug modification, drug without indication, ADR, inappropriate administration, drug interactions, overcoming drug class duplicate, conversion of intravenous to oral medications, non-compliance with guidelines, drug follow-up, and TDM. Literature also supports the pharmacist’s role in medication therapy management [45,46,47]. Drug information for healthcare professionals and patient counselling were among the pharmacists’ interventions identified in this review, which aligns with other studies [10,14,28,48,49]. These actions promote the safety of COVID-19 patients using anti-infectives, antibiotics, and anticoagulants. This shows pharmacists can play an important role in alleviating the burden on primary care providers by providing patient education in disease management [50,51,52]. According to Khatiwada et al., the COVID-19 pandemic has influenced people to seek medication-related information from pharmacists working in drug information centers and community pharmacies [28]. Included studies reported that aside from the traditional role of ensuring adequate drug supplies during the COVID-19 pandemic, pharmacist interventions focused on therapeutic dosage adjustment for COVID-19 affected special populations, such as chronic kidney disease patients, immunocompromised patients, and patients with comorbidities (HIV, hypertension, and diabetes). These results are also well supported by the studies of Wang et al. and Rodriguez et al. [42,53]. Review findings reported pharmacists regularly monitored the records of patients admitted to ICUs, reviewed their prescriptions and lab reports, managed their medication errors and did interventions to balance their electrolytes and these findings were supported by the literature [54,55,56,57,58,59].

According to Parez et al., pharmacists provided treatment to COVID-19 and non-COVID-19 patients to identify the medications that necessitate special care, particularly those engaged in the COVID-19 management [35]. The presence of pharmacists resulted in a significant reduction in drug-related prescription issues, particularly for antithrombotic and antibacterial drugs in the COVID-19 positive group, and instruction of patients on how and when to take medications [35]. One included study reported that community pharmacies with tele-pharmacy services could improve patient access to pharmacy care, particularly for COVID-19 patients, and reduce dispensing errors [37]. In some countries, hospital pharmacy services adopted tele-pharmacy for their outpatient consultation and drug dispensing services to optimize clinical outcomes and reduce the risk of contagion during the COVID-19 pandemic [51,60,61,62,63]. Community pharmacies acted as an information hub for the public, providing objective, unbiased and updated information on chronic diseases and their complications along with COVID-19, available management approaches for COVID-19, adverse effects of the medications and drug interactions using drug bulletins and pamphlets through telephonic communication or in-person [27,28,64]. Overall, the acceptance rate for the interventions offered by pharmacists was significant, indicating that pharmacists are thought to have extensive knowledge of drug-related issues and excellent communication abilities. However, additional information about physicians’ reasons for non-acceptance should be obtained to improve pharmacy services [65].

## 5. Further Research

In future, researchers should plan and develop studies with more rigorous methodologies, highlighting well-defined interventions that pharmacists could provide to encourage and guide the expanded role and activities that pharmacists could undertake in unforeseen pandemics, such as COVID-19. The roles of community pharmacists should be the main focus of further studies because they are the most accessible healthcare providers in the area, provide drive-thru and home delivery services, offer telehealth counselling and psychological support, and refer patients suspected of having COVID-19 to hospitals. Studies should also assess the impact of pharmacists’ role in the long-term outcomes of COVID-19 patients, such as QOL and mortality. The cost-effectiveness of such services and the research about the involvement of pharmacists in writing the prescription for the COVID-19 patients should be explored in future studies.

## 6. Strengths and Weaknesses

This is the first systematic review to evaluate the impact of pharmacist-delivered services on improving COVID-19 patients’ pharmaceutical care outcomes. COVID-19 being a new disease, the number of studies reporting the pharmacists’ role is less than anticipated. Therefore, the findings from this study are not generalizable in all settings. Even though there were fewer included studies, important interventions were reported. This review can serve as a guide for pharmacists in countries where the pharmacists did not have much involvement beyond the dispensing of medicines during the COVID-19 period. Moreover, using the DEPICT tool to characterize the intervention nature eliminated the rater effect. However, this study does have certain restrictions. No study examined patient health outcomes such QOL, adherence, morbidity, death, rehospitalization, or the economic impact of interventions. The number of areas of COVID-19 treatment and management is rapidly progressing. The included studies’ interventions only focused on services for COVID-19 patients. We could not find original research conducted about the potential roles of pharmacists in infection prevention and control, testing, and COVID-19 vaccinations.

## 7. Conclusions

Studies included in this SR described the beneficial impact of pharmacist-delivered services in the treatment of COVID-19, such as identifying drug regimen issues, dosage issues, wrong drug detection, managing drug interactions, optimizing therapy, TDM of antivirals, anticoagulants, antimicrobials, and patient use of medicines for other concurrent chronic health problems. Future research should evaluate pharmacist interventions concerning essential outcome measures, such as disease outcomes, mortality, quality of life, and hospital (re)admissions. Additionally, prescribing roles for pharmacists and economic evaluations of pharmacist interventions should be considered in future studies.

## Figures and Tables

**Figure 1 healthcare-10-01630-f001:**
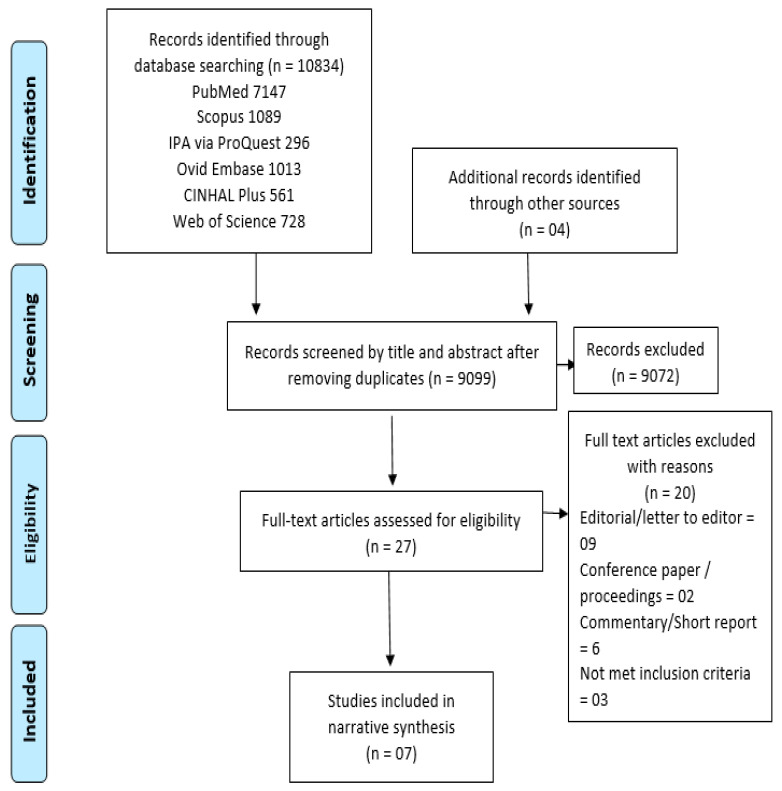
Flowchart of the study inclusion process.

**Table 1 healthcare-10-01630-t001:** Characteristics of included studies.

Author (year)	Objective	Study Design	Sample Size	Mean Age (years)	Study Duration	Country and Setting	Included Patient Characteristics	Outcomes Assessed	Quality of Studies total Score *
Perez et al., 2020	Comparison of clinical pharmacists’ interventions between two care groups COVID-19 positive and COVID-19 negative and to identify drugs that require special attention	Prospective cohort study	N = 438, COVID-19 positive status group: 222 COVID-19 negative status group: 216	COVID-19 positive group: 68.0 COVID-19 negative group: 69.0	1 month	Single centre, Bedside of Lille University Hospital, France	Patients admitted to COVID-19 units in the study hospital during the study duration, and those who benefitted from the pharmaceutical care were included in the study	Drug prescribing errors	8
Ibrahim et al., 2020	Differences in rates and types of pharmacist interventions related to COVID-19 and medication dispensing errors across community pharmacies with and without tele pharmacy services.	Prospective cohort study	N = 52, 26 Tele pharmacies (Test group), 26 traditional pharmacies (control group)	NR	1 month	Community pharmacies in all 7 states of the United Arab Emirates	Community pharmacies within the study locality that offered tele pharmacy services to COVID-19 patients vs. those that didn’t provide tele pharmacy.	Medication dispensing errors	7
Collins et al., 2020	To describe the institution’s strategy to deploy pharmacy resources and standardize pharmacy processes to optimize the management of patients with COVID-19.	Retrospective cohort study	N = 197	67 ± 16.7	Half month	537-bed teaching hospital located in Michigan, USA	All patients during the study period with a documented pharmacy intervention and a positive SARS-CoV-2 test were included in the analysis.	Optimize the management of patients with COVID-19 and quantify the volume and scope of pharmacist interventions.	5
Wang et al., 2021	Share professional experiences on medication optimization and provide a feasible reference for the pharmaceutical care of critically ill patients with COVID-19.	Retrospective cohort study	N = 22	66.3	Followed for the duration of inpatient stay in ICU	First Affiliated Hospital of Zhejiang University, China	Critically ill COVID-19 patients admitted to ICU for whom clinical pharmacists made medication recommendations.	Medication optimization	3
Alwhaibi et al., 2021	Assess types of interventions made and medication errors encountered by Pharmacists providing health care services to critically ill COVID-19 patients.	Retrospective cohort study	N = 79	58.8	Followed until patients are transferred to another ward/ discharged/died	Diriyah hospital in Riyadh, Saudi Arabia	Critically ill patients of COVID-19 admitted to ICU were included.	Identification of medical errors	5
Al-Quteimat et al., 2022	To study the role of the hospital pharmacists in the management of admitted patients with COVID-19 by analyzing the documented pharmacists’ clinical interventions and assess their type, rate, acceptance by physicians, clinical significance, and impact on overall patient care processes.	Retrospective cohort study	N = 202	46.3	4-month	360-bed tertiary care hospital in the United Arab Emirates, Abu Dhabi	Adult patients (age ≥ age 18) with confirmed COVID-19 diagnosis.	Clinical significance of pharmacist-initiated therapy optimization.	5

NR: Not reported, DRPs: Drug-related problems, CPOE: Computerized physician order entry, ICU: Intensive care units, MDEs: Medication dispensing errors. * 7–9 high quality, 4–6 high risk, and 0–3 very high risk of bias.

**Table 2 healthcare-10-01630-t002:** Description of pharmacist intervention according to DEPICT version 2.

Authors	Recipients	Mode of Contact with the Recipient	Setting Where Recipient Received Treatment	Methods of Communication	Clinical Data Sources	Classification of Intervention	Pharmacist Action(s)	Timings of Pharmacists Action	Frequency of Contacts	Materials that Support Action(s)	Changes in Therapy and Lab Tests Reported
Perez et al., 2020	COVID-19 Patients and physicians	One-to-one contact with Patients and Physicians	Hospital bedside	Written messages, Phone calls	Databases of patients’ medical history, medical prescriptions through EMR and CPOE.	Prescription analyses were performed as defined by the French Society of Clinical Pharmacy (SFPC).	Pharmaceutical care interventions (drug-related problems, non-conformity to guidelines, drug follow-up, under/overdosage, drug without indication, side effect, inappropriate administration, drug interactions).	Throughout the patient’s stay in the hospital.	Daily	EMR, written messages and phone calls	Yes
Surapat et al., 2020	COVID-19 Patients and physician	One-to-one with Patients and Physicians	Hospital bedside	Phone call or mobile chat application	Admission record, doctor’s orders, medication history, laboratory data, X-ray/CT scan reports, and progress notes.	Followed Thailand’s national guidelines	TDM, drug dose consultations, adverse effect monitoring	Throughout the patient’s stay in the hospital.	Daily	EMR	Yes
Ibrahim et al., 2020	COVID-19 Patient	One to one Face to face contact with patients.	Recipient’s home	Phone calls, Social media, video conferencing	NR	CDC guidelines and MDE classification	Pharmacies in the test group utilized the available IT tools to deliver remote pharmaceutical services like filling out prescriptions, medication reviews, patient counselling, and home. Delivery of medications to patients,	NR	NR	EMR	Yes
Collins et al., 2020	COVID-19 Patients and physicians and other care providers	One-to-one contact with a physician but no direct contact with patients.	Hospital bedside	Phone calls, instant messaging, and secure text messaging.	EMR	COVID-19 syndrome-specific intervention was developed.	Pharmaceutical care, optimization of medication therapy, streamlining of regimens for nursing workflow efficiency, and managing drug shortages.	Throughout patient admission	Daily	Written Documentati and EMR	Yes
Wang et al., 2021	COVID-19 Patients and physicians	One-to-one contact with patient and Physicians.	Hospital bedside	NR	Comprehensive medical history and multidisciplinary ward rounds.	Pharmaceutical Care Network Europe Foundation Classification V 9.0	Identify drug-related problems, and make medication recommendations.	Throughout patient admission	Daily two time	EMR	Yes
Alwhaibi et al., 2021	COVID-19 Patients	NR	Hospital bedside	NR	EMR	NR	Overcome drug class duplicate, missing drug, error in dosing regimen, in cases where medication is not available, reject drug order, re-order requested and non-privileged prescriber.	Throughout patient admission	Daily	EMR	Yes
Al-Quteimat et al., 2022	COVID-19 Patients and physicians	One-to-one contact with a physician	Hospital bedside	Epic^®^; (I-Vent) built in the electronic hospital health information system.	EMR	Local institutional guideline	Optimization of therapy, avoidance of adverse drug the events, improved communication, and cost savings.	Throughout patient admission	Daily	EMR	Yes

NR = Not reported, EMR = Electronic medical records, HCP = Health care provider, PI = Pharmacist intervention, TDM = Therapeutic drug monitoring, MDEs = Medication dispensing errors, CPOE = Computerized physician order entry, CDC = Centre for disease control, IT = Information technology.

**Table 3 healthcare-10-01630-t003:** Description of outcomes of pharmacist interventions reported in included studies.

Author	Number of Interventions Performed	Dosage Errors Identifications & Resolution	Adverse Drug Reactions	Drug Modification	Drug Interactions	Removal of Obsolete/Duplicate Medications	Miscellaneous OUTCOMES	Physician’s Level of Acceptance
Perez et al., 2020	—A total of 188 PIs were performed on the medication prescriptions of 118 patients: 64 and 54 interventions for positive and negative groups, respectively resulting in 1.6 Inter/patient.	—Incorrect dosage represented 36.7% (69/188) interventions: 27.9% (29/104) for the COVID-19-positive group and 47.6% (40/84) for the COVID-19-negative group. —Duplicate medication and non-adjustment of heparin to renal function) and concerned 24.4% (10/41) of PIs on antithrombotic: six and four PIs for positive and negative groups, respectively.	NR	—Antithrombotic agents (PIs = 20.7%, 39/188), antibacterial for systemic use (PIs = 13.8%, 26/188), and drugs for gastric acid-related disorders (PIs = 6.4%, 12/188) were modified.	NR	—The most frequent PI in 34% (64/188) of cases was terminating a drug: 27.9% (29/104) for the COVID-19- positive group and 47.6% (40/84) for the COVID-19- negative group.	—Second drug-related problem was the non-conformity with guidelines —No interventions were made for COVID-19-specific drugs like remdesivir. —Therapeutic care of patients with general anti-infective agents for systemic use represented 17.6% (33/188) of PIs in both groups. —PIs guided three patients about the appropriate use of inhaling devices when transferred to pneumology wards.	—COVID-19-positive patients were 88.5% (92/104). —COVID-19-negative patients were 90.5% (76/84).
Surapat et al., 2020	NR	NR	—PIs managed the common ADRs e.g., diarrhoea, body rashes, and induced hepatitis caused by protease inhibitors, azithromycin and tocilizumab.	NR	NR	NR	—PIs managed critically ill patients, including TDM. —Individualized drug dosing in special populations like chronic kidney disease and liver disease patients.	NR
Ibrahim et al., 2020	NR	—PIs in the test group reported (16.5%) wrong quantity errors. —Control group reported (20.7%) wrong quantity and (9.3%) wrong strength errors.	NR	—PIs optimized overdose in test group 296 (4.0%) and control group 66 (2.1%) —Optimized sub-therapeutic dose or duration in test group 166 (2.3%) and control group 41 (1.3%).	—PIs changed medication due to potential DDI or contraindication (potential allergy, pregnancy, etc.) in test group 1497 (20.4%) and control group 507 (16.4%).	—PIs removed duplicate drugs in test group 564 (7.7%) and control group 91 (2.9%)	—PIs detected errors in test group pharmacies were wrong patient (37.5%), followed by * wrong drug (20.3%). —Control group: wrong drug errors related PIs (40.9%).	NR
Collins et al., 2020	—A total of 1572 PIs were documented in 197 patients. —The average number of inter/patient were 8.	—Dosing adjustment represented 242 (15.4%) PIs.	—43 (2.7%) PIs avoided and managed Adverse drug events.	—250 (15.9%) PIs simplified the therapeutic regimen.	—Drug-drug interactions 32 (2%).	NR	—66.7% of interventions were done in ICU patients. —Prophylaxis 155 (9.9%) —Electrolytes 57 (3.6%) —Fluid management 50 (3.2%)—Nonspecific interventions (12.6%).	NR
Wang et al., 2021	—A total of 111 PIs were reported. —The average number of inter/patient were 5.04.	—Dose adjustment represented 17 (15.3%) PIs.	—58 (52.3%) PIs avoided and managed Adverse drug events.	— 22 (19.8%) PIs added a new drug to gain better therapeutic outcomes.	NR	—PIs did drug discontinuation in 35 (31.5%) interventions.	—64 PIs were related to antibiotics and antifungal drugs, 39/64 (60.9%) for treatment effectiveness and 25/64 (39.1%) for adverse drug events.	—The acceptance rate of PIs was 106 (95.5%).
Alwhaibi et al., 2021	—A total of 470 PIs were reported. —The average number of inter/patient were 5.9.	—151 (32%) PI solved the errors due to dosing adjustments. E.g., dose, duration, infusion rate, missing dose, missing information.	NR	NR	NR	—Drug class duplicate 55 (11.7%). For example, The patient is on Vancomycin and prescribed Linezolid.	—40.6% of PIs deal with the medication shortage of which 40.3% were substituted with alternative medications. —Most common pharmacological groups associated with interventions were antibiotics 16.8%, electrolytes/minerals 11.7%, and vitamins 9.4%.	NR
Al-Quteimat et al., 2022	—A total of 484 PIs were reported. —The average number of inter/patient were 2.4.	—Pharmacist did 26 (5.4%) interventions in dosage changes. —Pharmacist did 19 (3.9%) interventions to convert intravenous dose to oral dose.	—18% of the interventions resulted in preventing potential adverse drug reactions.	—12 (2.5%) interventions were done related to therapeutic modifications. —Admission medication reconciliation 26 (5.4%). —Drug therapy recommendations 14 (2.9%).	NR	—Pharmacist did 19 (3.9%) drug discontinuation interventions. —Resolving duplicate therapy 22 (4.5%).	—Pharmacists did (149, 30.8%) antibiotics stewardship interventions were, constituting 31.1% of the total interventions. —Overall, 50.8% (246) of the interventions rated “moderate” clinical significance using the clinical significance scoring tool. — “Optimized therapy” was the most commonly reported outcome (58.8%) of PIs.	—The physicians’ acceptance rate of was 94.7% (357 accepted out of 377 interventions included).

NR = Not reported, PI = Pharmacist interventions, HCP = Health care provider, TDM = Therapeutic drug monitoring, MDEs = Medication dispensing errors, CPOE = Computerized physician order entry, CDC = Centre for disease control, DDI = Drug-drug interactions, * wrong drug = The medicine dispensed differs from the medication listed on the prescription.

## Data Availability

The data generated or analyzed during this review are included in this published article (and Appendix A). Additional data are available from the corresponding author on reasonable request.

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
