# Peer review of "Interventions and Impact of Pharmacist-Delivered Services for People Infected with COVID-19: A Systematic Review"

_healthcare, 2022, doi:10.3390/healthcare10091630_

Round 1

Reviewer 1 Report

1. American Pharmacists Association is abbreviated as APhA (not APA).

2. The state of "Total 2825 interventions were made by pharmacists in inpatients." should say "A total of 2825 interventions were made by pharmacists in inpatients."

3. "Al quteimat" should be "Al-Quteimat."

4. Remove bullet points from table.

5. Under Strengths and Weaknesses, please change "couldn't" to "could not."

Author Response

Dear reviewer

Thank you for your constructive feedback.

Point 1. American Pharmacists Association is abbreviated as APhA (not APA).

Response 1: We have made the changes suggested by the reviewer. [Addressed in Page No. 4; Line No. 79]

Point 2. The state of "Total 2825 interventions were made by pharmacists in inpatients." should say "A total of 2825 interventions were made by pharmacists in inpatients."

Response 2: We have made the suggested changes. [Addressed in Page No. 10; Line No. 203]

Point 3. "Al quteimat" should be "Al-Quteimat."

Response 3: Thank you for highlighting this typos error. We have made a changes done as suggested. [Addressed in Page No.11; Line No. 227]

Point 4. Remove bullet points from table.

Response 4: We have now removed the bullets from table. [Addressed in Table 3]

Point 5. Under Strengths and Weaknesses, please change "couldn't" to "could not."

Response 5: We have made changes as suggested. [Addressed in Page No. 16 Line No. 333]

In addition to your suggestions, We have corrected spelling and typographical errors throughout the manuscript including tables. Furthermore, we have thoroughly updated the tables to improve readability and visualization.

Reviewer 2 Report

This manuscript is a systematic review regarding pharmacist interventions on the outcomes of COVID-19 patients. Literature search methods were sound, and the results of this review are of interest to healthcare professionals taking care of patients with COVID-19. Therefore, I can see the value of this manuscript published in this journal.

However, the submitted study has major concerns to be published in this journal as the current version. I would like have the authors address the following points:

1. Overall, several typographical and grammatical errors were found throughout the manuscript. I would highly recommend the authors proofread their manuscript and improve their writing.

2. The manuscript should be written based on the journal format. For example, legend for Figure 1 is missing, data availability statement is missing, abstract format appears in correct, etc.

3. Several outcomes of interests were listed, but which one was considered as the primary outcome? Please clarify. This will assist readers in determining how to generalize study findings to their real-world patients.

4. All tables included very detailed information. Although this is very helpful for readers extracting relevant information, tables were very hard to read. Authors should summarize all of the tables to improve visualization.

5. Authors have mentioned PICOS approach was used throughout their review process. Then, one of the tables should have separate headings to describe each of the PICOS, specifically the comparator or control group. It appears the patient population and the comparator were combined in one heading. Also, for the purpose of this review/study, patients with COVID receiving pharmacy services should be the testing group, and the control group should be those receiving standard of care without pharmacist-led services…? One of the studies included COVID-negative patients as their control group, and authors critique in this regard should be included in the discussion.

6. In order to evaluate the feasibility of implementing pharmacist-led services for patients with COVID-19 in the real world, the number of pharmacists, the number of interventions made (total and/or per pharmacist) with distributions for each type of intervention, qualification of the pharmacist, etc. should also be clarified and commented in the discussion section.

Author Response

Dear reviewer

Thank you for your constructive feedback.

This manuscript is a systematic review regarding pharmacist interventions on the outcomes of COVID-19 patients. Literature search methods were sound, and the results of this review are of interest to healthcare professionals taking care of patients with COVID-19. Therefore, I can see the value of this manuscript published in this journal. However, the submitted study has major concerns to be published in this journal as the current version. I would like have the authors address the following points:

Response: First, on behalf of coauthors, we would like to thank you for your encouraging comments. We have given point to point response to your suggestions and comments.

Point 1. Overall, several typographical and grammatical errors were found throughout the manuscript. I would highly recommend the authors proofread their manuscript and improve their writing.

Response 1: We have revised the manuscript and tried to improve the writing. Typographical and grammatical errors are also corrected. [Addressed throughout the manuscript highlighted in track changes]

Point 2. The manuscript should be written based on the journal format. For example, legend for Figure 1 is missing, data availability statement is missing, abstract format appears in correct, etc.

Response 2: Dear reviewer, we have revised the manuscript as per journal guidelines and included figure 1 legends and statements like data availability, funding, etc.    

Point 3. Several outcomes of interest were listed, but which one was considered as the primary outcome? Please clarify. This will assist readers in determining how to generalize study findings to their real-world patients.

Response 3: Thank you for your valuable comments and we agree with you that separating primary and secondary outcomes will assist readers in determining how to generalize study findings to their real-world patients. We have now clarified the primary and secondary outcomes. [Addressed in Page No. 07; Line No. 141-151]

Point 4. All tables included very detailed information. Although this is very helpful for readers extracting relevant information, tables were very hard to read. Authors should summarize all of the tables to improve visualization.

Response 4: Tables have been revised and summarized to improve visualization. [All tables 1-3]

Point 5. Authors have mentioned PICOS approach was used throughout their review process. Then, one of the tables should have separate headings to describe each of the PICOS, specifically the comparator or control group. It appears the patient population and the comparator were combined in one heading. Also, for the purpose of this review/study, patients with COVID receiving pharmacy services should be the testing group, and the control group should be those receiving standard of care without pharmacist-led services…? One of the studies included COVID-negative patients as their control group, and the authors critique in this regard should be included in the discussion.

Response 5: Dear reviewer, thank you for highlighting it. We have included the description of PICOS in the table attached in the supplementary file. [Addressed in Supplementary file Table 1S]

The changes made is as follows:  “According to Parez et al., pharmacists provided the treatments to COVID-19 and non-COVID-19 patients in order to identify the medications that necessitate special care, particularly those engaged in COVID-19 management. The presence of pharmacists resulted in a significant reduction in drug-related prescription issues, particularly for antithrombotics and antibacterial drugs in the COVID-19 positive group, and also instructed patients on how and when to take medications.” [Addressed in Page No. 14; Line No. 292-296]

Point 6. In order to evaluate the feasibility of implementing pharmacist-led services for patients with COVID-19 in the real world, the number of pharmacists, the number of interventions made (total and/or per pharmacist) with distributions for each type of intervention, qualification of the pharmacist, etc. should also be clarified and commented in the discussion section.

Response 6: Dear reviewer we have added further details in the discussion section. Overall, we have revised the discussion section throughout. [Addressed on Page No. 12-13 Line No. 259-269]

Following changes are made in the discussion:

“In all studies, pharmacists were graduated, licensed, and trained in infectious diseases, and their knowledge of COVID-19 was continuously updated. Pharmacists performed 2,825 interventions, and physicians accepted more than 90% of the interventions. Pharmacists collected medication-related issues from patient files and online records, analysed them, and then provided educational materials and solutions to patients and physicians. Since gatherings were not allowed because of the pandemic’s safety measures, educational materials/interventions were delivered to the physician in form of pamphlets, and posters with all kinds of observed medication errors hung in various departments and prescribers’ rooms. On-site education and verbal communication through phone calls were carried out. Moreover, online educational meetings for sharing experiences, online weekly educational meetings with clinicians, and the distribution of clinical protocols and guidelines were also done."

We have also corrected spelling and typographical errors throughout the manuscript including tables. Furthermore, we have thoroughly updated the tables to improve readability and visualization.

Reviewer 3 Report

Ahmed et al. reported the role of pharmacist’s intervention in covid-19 patient management. In general, the article is well written, but the major concern I have is that 6 studies are too less of a sample size to predict how significant the outcome is. On the other hand, it’s understandable that a pharmacist’s role is expected and necessary for better management of any treatment approach for any disease condition. Hence the same is expected for covid-19 management. I am not sure if the reported study provides any new and significant information which can also be useful. I am sorry that my review could not be positive.

Minor comment:

The tables are very difficult to follow. There are a lot of mistakes and errors in spelling in table 3. 

Author Response

Dear reviewer

Thank you for your feedback.

Point: Ahmed et al. reported the role of pharmacist’s intervention in covid-19 patient management. In general, the article is well written, but the major concern I have is that 6 studies are too less of a sample size to predict how significant the outcome is. On the other hand, it’s understandable that a pharmacist’s role is expected and necessary for better management of any treatment approach for any disease condition. Hence the same is expected for covid-19 management. I am not sure if the reported study provides any new and significant information which can also be useful. I am sorry that my review could not be positive.

Minor comment:

The tables are very difficult to follow. There are a lot of mistakes and errors in spelling in table 3.

Response 1: Dear reviewer, thank you for the feedback. Correction there are 07 studies not 06 studies. While we acknowledge, that the number of articles found included in our study is less, we believe the findings have unearthed new opportunities for pharmacists, especially during the health emergencies like COVID-19 pandemic. While the roles of pharmacists might seem redundant in many developed countries, we believe this is not the case everywhere, especially in developing countries. We believe this systematic review will serve as a guide to pharmacists from such countries. We incorporated your suggestion by adding this as a limitation of our study. Following additions in the ‘Strengths and Weaknesses’ section highlights this: 

“COVID-19 being a new disease, the number of studies reporting the pharmacists' role is less than the anticipation. Therefore, the findings from this study are not generalizable in all of the settings. Even though there were fewer included studies, important interventions were reported. This review can hence serve as a guide for the pharmacists in the countries where the pharmacists did not have much involvement beyond the dispensing of the medicines during the COVID-19 period.” [Addressed on Page No. 15 Line No. 323-328]

Thank you for your comments about the Tables. We have corrected spelling and typographical errors throughout the manuscript including tables. Furthermore, we have thoroughly updated the tables to improve readability and visualization.

Round 2

Reviewer 3 Report

Thanks for addressing my comments.